# Influence of Topiramate on the Synaptic Endings of the Temporal Lobe Neocortex in an Experimental Model of Hyperthermia-Induced Seizures: An Ultrastructural Study

**DOI:** 10.3390/brainsci11111433

**Published:** 2021-10-28

**Authors:** Piotr Sobaniec, Joanna Maria Lotowska, Maria Elzbieta Sobaniec-Lotowska, Milena Zochowska-Sobaniec

**Affiliations:** 1Department of Pediatric Neurology and Rehabilitation, Faculty of Health Sciences, Medical University of Bialystok, 15-274 Białystok, Poland; piotr.sobaniec@neuromaster.pl; 2Department of Medical Pathomorphology, Faculty of Medicine with the Division of Dentistry and Division of Medical Education in English, Medical University of Bialystok, 15-269 Białystok, Poland; joanna.lotowska@umb.edu.pl; 3Department of Pediatrics, Gastroenterology, Hepatology, Nutrition and Allergology, Faculty of Medicine with the Division of Dentistry and Division of Medical Education in English, Medical University of Bialystok, 15-274 Białystok, Poland; milena@neuromaster.pl

**Keywords:** hyperthermia-induced seizures, experimental febrile seizures, topiramate, ultrastructure of synaptic endings, morphometric analysis, temporal lobe neocortex, neuroprotection

## Abstract

The objective of this pioneering study was to assess potentially neuroprotective properties of topiramate (TPM), a broad spectrum and newer-generation antiepileptic used against damage to synaptic endings of the temporal lobe neocortex in experimental hyperthermia-induced seizures (HS). TPM (80 mg/kg b.m.) was administered in young male Wistar rats with an intragastric tube before and immediately after HS. Specimens (1 mm^3^) collected from the neocortex, fixed via transcardial perfusion with paraformaldehyde and glutaraldehyde solution, were routinely processed for transmission-electron microscopic study, i.e., for descriptive and morphometric analysis. The ultrastructure of neocortical neuropil components affected by hyperthermic stress showed distinct swelling of pre and post-synaptic axodendritic and axospinal endings, including total disintegration. Mitochondria were markedly damaged in synaptic structures. Axoplasm of presynaptic boutons contained a decreased number of synaptic vesicles. Synaptic junctions showed active zone-shortening. Preventive administration of TPM before HS induction demonstrated neuroprotective effects against synaptic damage in approximately 1/4 of these structures. Interestingly, beneficial effects on synapsis morphology were more common in perivascular zones close to well-preserved capillaries. They were demonstrated by smaller swelling of both presynaptic and postsynaptic parts, well-preserved mitochondria, an increased number and regular distribution of synaptic vesicles within axoplasm, and a significantly increased synaptic active zones. However, topiramate used directly after HS was ineffective in the prevention of hyperthermia-evoked synaptic injury. Our findings support the hypothesis that topiramate applied before HS can protect some neocortical synapses via the vascular factor by enhancing blood–brain barrier components and improving the blood supply of gray matter in the temporal lobe, which may be significant in febrile seizure-prevention in children.

## 1. Introduction

Febrile seizures are still the most common neurologic disorders in the pediatric age group. Hyperthermia-induced seizures (HS), also called syn. hyperthermic convulsions, in rats have been used as a model of febrile seizures in children. A high percentage, i.e., approximately 30–40% of the population, with fever-induced seizures will have a recurrence during early childhood. It is reported that even though simple febrile seizures are usually benign, children with complex FS are at risk of subsequent epilepsy later in life, including, e.g., drug resistance epilepsy of the temporal lobe associated with mesial temporal sclerosis [1,2,3,4,5,6,7,8,9]. It has been emphasized in literature that temporal lobe epilepsy (TLE) and hippocampal mesial temporal sclerosis commonly arise following early life-long seizures and especially after febrile status epilepticus [3,5,6,7,10,11,12]. Worthy of note is that TLE is the most prevalent type of epilepsy, although its origin is still poorly understood. This pathology mainly due to its frequent drug-resistance remains a huge challenge for a pediatric neurologist [1,2,3,4,6,7,11].

Therefore, it seems urgent to conduct in-depth studies on the pathomorphogenesis of microscopic changes underlying injuries induced by recurrent hyperthermic convulsions in various morphological structures of the CNS observed in an experimental model of HS that would follow the methodical criteria corresponding to this form of fever-induced seizures in pediatric patients. Such a rat model of hyperthermic convulsions comparable to pediatric FS was elaborated on years ago in our center [13].

The last two decades have brought on an intensive search for pharmacological neuroprotection against various damaging factors of the CNS, including prolonged and recurrent FS in children.

Topiramate (TPM) is a commonly used antiepileptic and newer-generation drug with multi-faced pharmacological action as well as a low potential of serious side effects [14,15,16,17,18,19,20,21]. It is an effective, broad-spectrum anticonvulsant medication used in the treatment of various types of epilepsy, in both partial and generalized seizures [14,15,17,18,19,21,22], and in the prevention of recurrent FS [16,23] in migraine prevention [24,25], as well as lately in pharmacotherapy for alcohol abuse and other addictions [26]. However, it should be emphasized that although neurochemical studies conducted on TPM in various animal models indicate its multidirectional effects on the CNS, mainly through the action on GABA and the inhibition of the excitatory amino acids [20,27,28,29,30,31,32,33], its exact mechanism at the ultrastructural level still remains unclear.

Therefore, the aim of the study was to estimate potentially neuroprotective effects of TPM against submicroscopic damage to the synaptic endings in the temporal lobe neocortex caused by hyperthermic convulsions in rats using ultrastructural descriptive and morphometric analysis.

The current work is a continuation of our histological, morphometric, and electron microscopic studies on the effects of this antiepileptic drug on the selected structural CNS components, mainly the ammonal cortex, specifically the blood–brain barrier (BBB), pyramidal neurons, and astrocytes in the HS model, as well as a continuation of our recent research into the synaptic endings of the hippocampal CA1 and CA3 sectors [13,34,35,36,37].

## 2. Materials and Methods

### 2.1. Animals

The retrospective electron-microscopic analysis of the effect of TPM on the chosen morphological structures of the rat brain, including neuropil components of the temporal lobe neocortex, in the experimental model of HS was conducted at the Department of Medical Pathomorphology, Medical University of Białystok.

The experiment used 18 young male Wistar rats aged 22–30 days. We used only male rats because in children, febrile seizures are more common in boys, as the male to female ratio is approximately 1.6 to 1.0 [2]. The rats were divided into four groups: three experimental and one control (five rats in each experimental group and three in the control). The animals were pre-selected according to the standard pharmacological screening tests. All procedures were performed in strict accordance with Helsinki Convention Guidelines for the care and use of laboratory animals. The study was approved by the Ethical Committee of our university.

### 2.2. Model of Febrile Seizures

The HS group included rats with induced febrile seizures. Hyperthermic stress was induced by placing the animals in 30 × 30 × 60 cm water baths filled with 45 °C of warm water. Water temperature was maintained at the same level. The rats were placed in water for 4 min until convulsions appeared and then moved to a separate container lined with lignin. The animals, except for the controls, were put in the warm water bath for four consecutive days. Most rats subjected to hyperthermia hot water developed rapidly myoclonic jerks and then generalized seizures, with vigorous shaking of the head, ears, and upper and lower limbs, and an especially violent vibration of the tail.

In the HS + TPM group, topiramate (Topamax, f. Jaansen-Cilag; 80 mg/kg b.m. dissolved in 2 mL of normal saline) was administered with an intragastric tube immediately after each convulsion episode (every rat received the drug four times).

In the TPM + HS group, topiramate was applied in the same way and at the same dose prior to the induction of febrile convulsions, i.e., 90 min before the animals were placed in the water bath.

Control animals and the HS group received only normal saline. A detailed description of the methodology has been described in our earlier paper [13,38].

Taking into consideration the several-times-faster metabolism of rats compared to that of humans, a dose of 80 mg/kg b.m. of TPM, in our opinion, is safe and comparable to the dose administered, e.g., to patients with drug-resistant epilepsy. Possibly, the use of a dose higher than 80 mg/kg b.m. might ensure more beneficial results and, at the same time, safe neuroprotective effects of the drug on the synaptic endings of the CNS structures studied. This, however, would require further neuropharmacological research extended with ultrastructural investigations.

### 2.3. Preparation for Transmission Electron Microscopy (TEM)

Seventy-two hours after the last convulsion episode, the rats were anaesthetized with Nembutal (25 mg/kg b.m., i.p.). Then, they were perfused intravitally and transcardially (through the left heart chamber to the superior aorta, with simultaneous clamping of the descending aorta and incision of the right atrium) with a fixative solution (approximately 200 mL/animal) containing 2% paraformaldehyde (f. Sigma) and 2.5% glutaraldehyde (f. Serva) in 0.1 M cacodylate buffer (f. Serva), pH 7.4, at 4 °C under pressure of 80–100 mmHg. After removal of the brains, temporal lobe samples were taken and fixed in the same solution for 24 h. Post-fixation was completed with 1% osmium tetroxide (OsO4) (f. Serva) in 0.1 M cacodylate buffer, pH 7.4, for 1 h. After dehydration in ethanol and propylene oxide (f. Serva), small specimens (1 mm^3^) of the temporal lobe neocortex were processed routinely for embedding in Epon 812 (f. Serva) and sectioned on a Reichert ultramicrotome (Reichert Ultracut S) to obtain semithin sections. The semithin sections were stained with 1% methylene blue (f. POCH) in 1% sodium borate (f. POCH) and preliminarily examined under a light microscope to select Epon blocks. Ultrathin sections were double-stained with uranyl acetate (f. Serva) and lead citrate (f. Serva), and examined with a transmission electron microscope (Opton EM 900, Zeiss, Oberkochen, Germany) as well as photographed with a TRS camera (CCD: camera for TEM 2K inside). The material obtained from the neocortex of the temporal lobe in the control group was processed using the same techniques as for the experimental groups.

### 2.4. Measurement of Neuronal Synapses and Quantitative Analysis

Fifty images of Gray type I synapses randomly selected and observed under 30,000× magnification were taken from each study group. The ImageJ software was used to measure the following parameters: postsynaptic density thickness (at the thickest part), active zone width, and synaptic cleft width (mean of the three values comprising the largest, intermediate, and smallest parts). The methodology of the morphometric study was prepared based on Guillery [39], Jjang et al. [40], Zhou et al. [41], and Han et al. [42].

### 2.5. Statistical Analysis

Data were analyzed by means of Statistica (StatView) statistical analysis software using the Kruskal–Wallis H-test. Values were presented as median values and interquartile ranges, and statistical significance was set at *p* < 0.05.

## 3. Results

### 3.1. Decriptive Ultrastructural Study

#### 3.1.1. HS Group

The ultrastructural changes observed in the synaptic endings of the temporal lobe neocortex in all HS rats, as compared to the control (Figure 1), were markedly enhanced in these neuropil components (Figure 2A–D).

Degenerating synaptic endings showed a whole spectrum of morphological features, from severe damage to necrosis. The extensive and severe damage to these structures was accompanied by the distinct injury of other neuropil components, i.e., dendritic processes of nerve cells and glial processes lying loosely in the neuropil.

Advanced submicroscopic alterations referred both to the axodendritic and axospinal synapses. The most pronounced was the substantial swelling of the pre and post-synaptic parts of synapses in the gray matter of the temporal lobe (Figure 2A–D). The cytoplasm of the enlarged terminal axons, i.e., axonal end-bulbs, and the cytoplasm of dendritic endings of the neocortex contained extensive, optically almost-empty fields or were filled with very fine residual microfibrillar material, which probably remained after the degenerated cytoskeletal elements dilated the smooth endoplasmic reticulum with vacuolar, electron-lucent structures (Figure 2A–D). Numerous vacuolar structures appearing within the cytoplasm of the synaptic endings led to their vacuolization.

The markedly degenerated synaptic endings frequently showed substantially damaged mitochondria. Mitochondrial abnormalities were characterized by their considerable swelling, large polymorphism, damage to mitochondrial membranes (Figure 2C,D), and eventual disintegration. The partially emptied mitochondrial matrix showed a deposition of microfibrillar substance (Figure 2D).

The ultrastructural picture of terminal axons revealed that the axoplasm of the presynaptic bulbs contained a substantially decreased number of synaptic vesicles and showed their abnormal distribution. Frequently, residual presynaptic vesicles accumulated abnormally in the form of clumpings mainly in the vicinity of the synaptic cleft or were unevenly dispersed throughout the axoplasm (Figure 2A–D). The cytoplasm of some degenerated synaptic endings, especially in their postsynaptic parts, showed fine, irregularly shaped heterogeneous electron-dense structures of mixed-electron density. Such structures were observed in the endoplasmic reticulum among cytoskeletal elements and in the mitochondrial matrix of some damaged and disrupted mitochondria. They may constitute the morphological feature of necrotic changes or apoptosis, corresponding to neuronal cell death due to hyperthermic stress (Figure 2A,B,D). However, the exact identification of such structures with mixed-electron density, as sometimes observed within the synapses, requires further in-depth studies, including immunological studies.

The synaptic junctions affected by hyperthermic stress showed a distinctly reduced length of the synaptic active zone and a decreased curvature of the synaptic interface; however, the synaptic cleft was quite frequently dilated and the postsynaptic part displayed a reduced thickness of the postsynaptic density (Figure 2A–C).

A distinct focal disintegration was commonly observed within the synaptic endings of the studied neocortex in animals with induced HS, which was demonstrated in the ultrastructure by the contours of shadows of the former synaptic structures.

#### 3.1.2. HS + TPM Group (the Antiepileptic Administered Immediately after the Induction of Febrile Seizures)

The electron-microscopic picture of the synaptic endings in the temporal lobe neocortex in animals, which immediately received TPM after hyperthermic injury, was similar to that observed in the HS group. It did not differ significantly either in quality or quantity from that obtained of rats who had undergone hyperthermic stress. Severe damage to the synaptic endings was persistent, specifically regarding their substantial swelling, mitochondrial injury, and markedly reduced length of the synaptic active zone (Figure 3A,B).

In this group we failed to find any neuroprotective effect of topiramate applied after neocortex synaptic damage induced by febrile seizures.

#### 3.1.3. TPM + HS Group (the Antiepileptic Administered Prior to Febrile Seizures)

In this experimental group, in the majority of observations, no distinct favorable effect was found on the ultrastructure of the neuropil components of the temporal lobe neocortex in animals receiving topiramate at a dose of 80 mg/kg b.m. before the induction of hyperthermic stress. Approximately 75% of the synaptic endings did not differ significantly from those observed in the HS group, showing features of distinct swelling, including necrosis. Within the cytoplasm, the majority of such terminal axons contained a markedly reduced number of synaptic vesicles. The synaptic active zone was reduced and the synaptic cleft was dilated.

The remaining 25% of the synaptic endings of the neocortex showed morphological features indicating marked neuroprotective effects of topiramate against synaptic damage. Interestingly, the favorable effect of TPM, administered preventively prior to hyperthermic stress induction, was mainly observed in the endings that were situated directly in the perivascular zones of relatively well-preserved capillaries or in the vicinity of such zones, which can be seen in Figure 4A,B.

The neuroprotective effect of the antiepileptic was manifested by less substantial swelling of the synaptic endings both in the pre and post-synaptic parts, with well-preserved mitochondria and granular endoplasmic reticulum channels (Figure 4A–D). We also observed an increase in the number of synaptic vesicles and their regular distribution within the axoplasm of the presynaptic end-bulbs (Figure 4A–D and Figure 5C,D).

Moreover, TPM administered prior to HS caused a significant lengthening of the synaptic active zone and synaptic interface curvature, as well as a thickening of postsynaptic density (shown also in Figure 4A–D and Figure 5C,D). Sometimes, although more seldom, the preventive application of the antiepileptic resulted in shorter synaptic clefts.

Interestingly, the findings of the ultrastructural picture (qualitative morphological analysis) fully corresponded to the results of the quantitative investigations of the analyzed parameters of the synaptic joints contained in Figure 6.

### 3.2. Morphometric Analysis

Figure 6 presents the measurements of postsynaptic density thickness (A), synaptic active zone length (B), and synaptic cleft size (C) in the control group (CG), hyperthermic-induced seizure group (HS), HS group with administrated topiramate (HS + TPM), and group with TPM administration before HS induction (TPM + HS).

## 4. Discussion

The current study is the first in neuropathology to document the effect of TPM applied in a dose of 80 mg/kg b.m. on the ultrastructural picture, including descriptive and quantitative assessment of synapses in the neocortex in an experimental model of HS.

In the study of HS as an experimental model of febrile seizures in children, elaborated in our center, brain maturity of young male Wistar rats aged 22–30 days is an equivalent to 3–5 years in humans [41,43,44].

At the electron microscopy level in the group of rats affected by repeated convulsions (HS group), we found a whole spectrum of pathomorphological features from the advanced injury of the synaptic endings to their necrosis. Most common was distinct swelling of pre and post-synaptic axodendritic and axospinal endings, as well as their total disintegration. Enlarged terminal axonal and dendritic endings contained almost-empty fields with fine residual microfibrillar material and markedly injured mitochondria. The axoplasm of the presynaptic boutons showed a distinctly decreased number of synaptic vesicles and an abnormal accumulation. The synaptic junctions showed a substantially reduced length of the synaptic active zone, increases in the synaptic cleft distance, and decreases in the postsynaptic density thickness in rats with HS, which was demonstrated in the conducted morphometric studies of the analyzed synaptic structures as compared to the control group (Figure 6).

However, TPM administered “prophylactically”, i.e., prior to HS (TPM + HS group), such as in the hippocampal CA1 and CA3 sectors in the analogous model of seizures [34], demonstrated a neuroprotective effect against hyperthermia-induced synaptic damage in the neocortex in approximately 1/4 of these structures, causing distinct improvement in the morphologic synaptic parameters. The neuroprotective effect of topiramate on the ultrastructural picture of the synapses was demonstrated mainly by less substantial swelling of the synaptic endings. Their cytoplasm showed well preserved mitochondria. An increase was noted in the number of synaptic vesicles and in their regular distribution within the axoplasm of the presynaptic parts of synapses.

A marked improvement was also observed in morphometric synaptic parameters, which was demonstrated by a significantly increased length of the synaptic active zone, a reduced dilation of the synaptic cleft size, and an increased thickening of postsynaptic density (Figure 6).

Interestingly, in our earlier submicroscopic study in an analogous experimental model of hyperthermia-induced seizures, when TPM was applied prior to induction of HS, the drug was found to have a beneficial effect on the structural status of approximately 1/3 of the population of protoplasmic astroglial cells of the hippocampal cortex and neocortex of the temporal lobe, which may be explained, among others reasons, by a protective action of cerebral cortex microcirculation [35].

It is worth adding that in light of recent research, this is most probably due to the effect of the drug at the astrocytes in the tripartite synapse where aquaporin 4 (AQP4) channels, which are mainly expressed in perivascular astrocyte endfeet, have been shown to play a major role in cell swelling and in the pathophysiology of epilepsy [45,46].

The population of glial cells, particularly astrocytes, appears to play critical and interactive roles in swelling and functional recovery [46,47,48], and are important targets for CNS disorders [46,48].

Recently, it was reported that topiramate may modulate the bicarbonate-driven pHi-regulation [49], which may have a beneficial effect on the synapse morphology and neurons of the neocortex when the drug is administered “prophylactically” before experimental HS.

Worthy of note is that TPM administered immediately after the induction of hypethermic stress had no beneficial effect on the structure of the synaptic endings, similarly to those in the ammonal cortex [34], which is reflected in the analyzed quantitative results.

It has been proved that topiramate has several mechanisms that may contribute to its anticonvulsant and neuroprotective properties, including antagonistic effects on ionotropic glutamate receptors of the AMPA/kainate and activation of GABAA receptors, which play an essential role in the inhibition of the excitotoxic neuronal damage [27,29,30,31,32,50]. A vital role in this mechanism has been ascribed to ion channels, i.e., voltage-gated Ca^2+^ channels located on the surface of the synaptic endings that are involved in neurotransmitter release. Therefore, TPM that antagonizes excitotoxicity by the inhibition of Ca^2+^ channel function in neurons is a particularly attractive target for neuroprotection [20,32,51,52].

Interestingly, in the current submicroscopic research, the neuroprotective effect of topiramate used as a “prophylaxis” was mainly found in the synaptic endings located directly in the perivascular zone of well-preserved capillaries or in the vicinity of such zones, which may indicate a crucial effect of the vascular factor in the morphogenesis of the neuroprotective action of this drug. This is in line with our previous ultrastructural studies on the morphological picture of the structural components of the BBB in the ammonal cortex in an analogous experimental model, in which TPM administered prior to FS at the same dose, i.e., 80 mb/kg b.m., caused distinct normalization of the morphological picture in over half of the capillaries of the CA1 and CA3 sectors [35], as well as a beneficial effect on the synaptic endings adhering in these sectors to well-preserved capillaries [34]. The protective effect of topiramate on the BBB components, mainly on the endothelial lining of capillaries, has been confirmed by electron microscopic studies of other authors who also observed a beneficial influence of TPM-pretreatment on the ultrastructural condition of the BBB components in experimental hyperthermic convulsions in rats [53]. A similar neuroprotective effect of topiramate against hyperthermia-induced synaptic damage was observed by Zhou et al., in the hippocampal CA1 area in immature rats with repeated FS after high-dose administration of a metabolic intermediate, i.e., fructose-1,6-diphoshate [41].

In our opinion, the favorable effect of the antiepileptic administered before FS on certain synapses of the temporal lobe neocortex observed in the current study, similar to in the previously assessed synaptic endings in the hippocampal CA1 and CA3 sectors, and in line with the assessment of other authors [53], can be associated with a considerable improvement in the CNS microcirculation, which results in a substantial enhancement of the BBB and thus better blood supply in the CNS.

It should be emphasized that the stabilization of BBB has recently been shown to improve the glymphatic function and hence functional recovery after conditions such as epilepsy [46,47]. Therefore, it cannot be ruled out that an analogous or similar mechanism of the BBB stabilization could have also taken place in our animal model of hyperthermia-induced seizures and, with the use of TPM, requires further in-depth research.

It should be added that the dose of topiramate applied in our HS model (80 mg/kg b.m.) as a neuroprotective dose [13], chosen according to literature references, has been used in various models of experimental seizures. For instance, it was administered, similar to our research, to young rats in hyperthermia-induced seizures [23,53], to adult rats in experimental status epilepticus [54], and in experimental temporal-lobe epilepsy [15]. Additionally, various doses of TPM (70 and 100 mg/kg b.m.) were used in adult rats against methylphenidate-induced oxidative stress and inflammation in the rat-isolated hippocampus [29] and in the rat-isolated amygdala [31].

The obtained results of the ultrastructural research, as the first in neuropathology, may in the future constitute an interesting comparative material (reference point) for similar morphological observations, including both descriptive and quantitative, carried out by other centers addressing the influence of antiepileptics on the CNS in experimental hyperthermic convulsions.

## 5. Conclusions

The current submicroscopic study, which is the first in neuropathology, showed that TPM used as a potentially preventive agent before experimental HS had, similar to the case of the previously assessed hippocampal CA1 and CA3 sectors, a distinct beneficial effect on approximately ¼ of the synaptic endings of the temporal lobe neocortex against hyperthermia-induced synaptic damage. In turn, the antiepileptic administered directly after HS was ineffective in the prevention of hyperthermia-evoked synaptic injury. The findings, both the current and the previous ones, support the hypothesis that the “prophylactic” application of TPM can protect the rat brain from structural damage induced by frequent, repetitive HS. In our opinion, the drug administered before a seizure occurs can act as a neuroprotective agent on certain synaptic endings, mainly on the intermediate pathway, which was expressed by the improvement in the morphological status of the BBB and better blood supply in the examined region of the CNS. Our results may have practical implications in the prevention of the effects of prolonged and recurrent hyperthermic convulsions in pediatric patients, which, however, still requires further neuropathological research.

## Figures and Tables

**Figure 1 brainsci-11-01433-f001:**
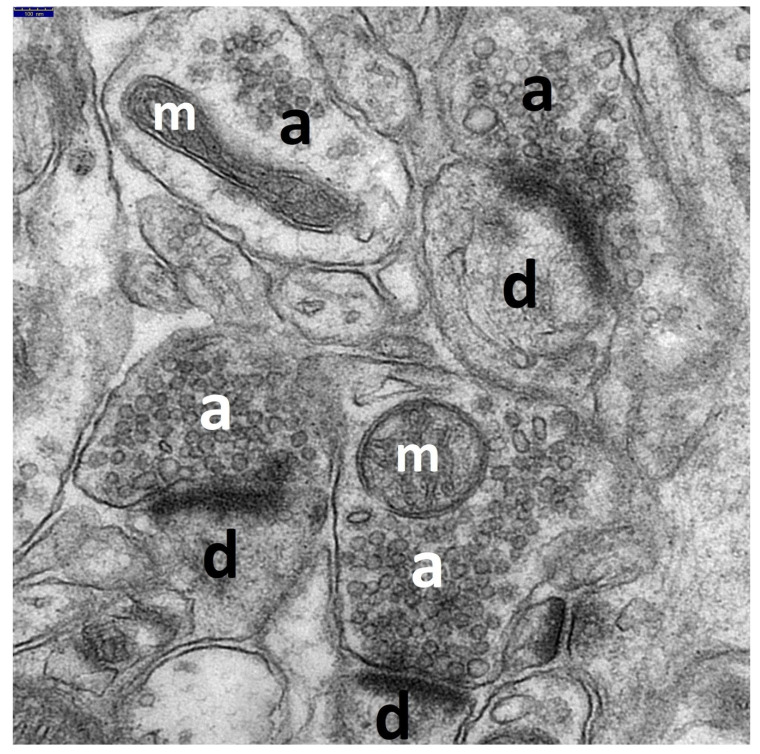
The electronogram demonstrates well-preserved neuropil components of the temporal lobe neocortex in the control group. The majority of the synaptic endings and axodendritic junctions show normal ultrastructures. The presynaptic (a) and postsynaptic (d) parts of the synapses are clearly seen. Most of the axonal-end bulbs are filled with a large number of synaptic vesicles, mainly spherical in shape, evenly distributed within the axoplasm, and containing unchanged mitochondria (m); one axonal ending shows a smaller content of synaptic vesicles. Normal picture of axodenditic junctions; synaptic active zones are distinct and long with well-preserved postsynaptic density thickness. Scale bar: 100 nm.

**Figure 2 brainsci-11-01433-f002:**
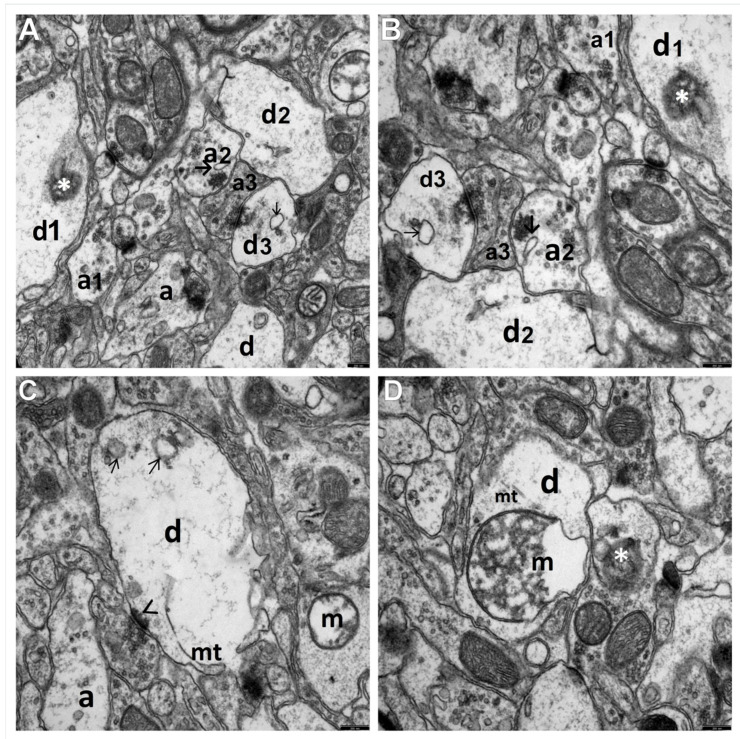
The electronograms show marked swelling and damage to the synaptic endings in the temporal lobe neocortex in the HS group. (**A**,**B**) Most visible is a group of substantially enlarged synapses; specifically, the presynaptic (a1, a2, and a3) and postsynaptic (d1, d2, and d3) parts of the synapses are remarkably swollen, filled with very fine microfibrillar material, and showing features of disintegration in places. Markedly reduced content of synaptic vesicles and their abnormal clumping within the axoplasm in the active synaptic region. In some dilated synaptic endings, fine vacuolar structures can be seen (→). The dendroplasm of the dilated synaptic ending shows the presence of an irregularly shaped structure with a heterogeneous electron density (*) that may correspond to a mitochondrion undergoing necrosis. Markedly reduced length of the synaptic active zones. (**C**) The electronogram demonstrates morphological details of a centrally located and markedly swollen postsynaptic part of the synapse (d). Within the dendroplasm there are optically almost-empty fields with residual microfilament elements of the cytoskeleton; in the upper part of this dendritic ending, transverse sections through single channels of the granular endoplasmic reticulum are visible (→); the lower part of the ending shows single microtubules (mt) with a markedly reduced length of the active zone (>). The majority of the surrounding neuropil components, mainly axonal-end bulbs (a), are swollen and contain a reduced number of synaptic vesicles. The adjacent swollen synaptic ending shows markedly swollen mitochondrion (m). (**D**) An interesting ultrastructural picture presenting a markedly swollen dendritic ending containing a substantially enlarged, involving most of the dendroplasm, and degenerated mitochondrion (m) situated in the center of the electronogram. Most of the mitochondrial matrix is filled up with merging-together fragments of disintegrating mitochondrial cristae that form a distinct relatively homogeneous microfibrillar substance; visible is also a large fragment of the emptied mitochondrial matrix. Above the mitochondrion, a residual number of microtubules (mt) can be seen. The neuropil component surrounding this dendritic ending, especially the axonal-end bulbs, is swollen with a reduced quantity of the synaptic vesicles; the cytoplasm of one of the synaptic endings shows a structure with mixed electron density (*) that may correspond to a degenerated and disintegrating mitochondrion. Scale bar: 250 nm (**A**–**D**).

**Figure 3 brainsci-11-01433-f003:**
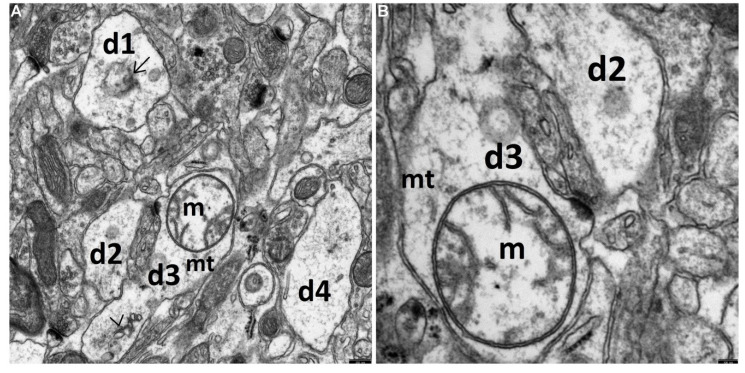
The cytoplasm of numerous synaptic endings of the temporal lobe neocortex is markedly swollen, without distinct features of the protective effect of the drug on the ultrastructure of these endings in the HS + TPM group. (**A**) The picture of degenerated, markedly enlarged synaptic endings, mainly dendritic ones (d1, d2, d3, and d4), containing the microfilament elements of the cytoskeleton. The cytoplasm of the centrally located dendritic ending (d3) contains markedly swollen mitochondrion (m) with peripherally arranged fragments of cristae within the mitochondrial matrix; transverse sections through dilated channels of the granular endoplasmic reticulum (>); and single residual microtubules (mt). The cytoplasm of the dilated postsynaptic part of the synapse (d1) located in the upper part of the figure contains a disintegrating structure, presumably a mitochondrion (→). The synaptic active zones are substantially shortened. A greater magnification (**B**) well demonstrates morphological details visible within the dendroplasm of the centrally located synaptic ending (d3). Scale bar: 250 nm (**A**) and 100 nm (**B**).

**Figure 4 brainsci-11-01433-f004:**
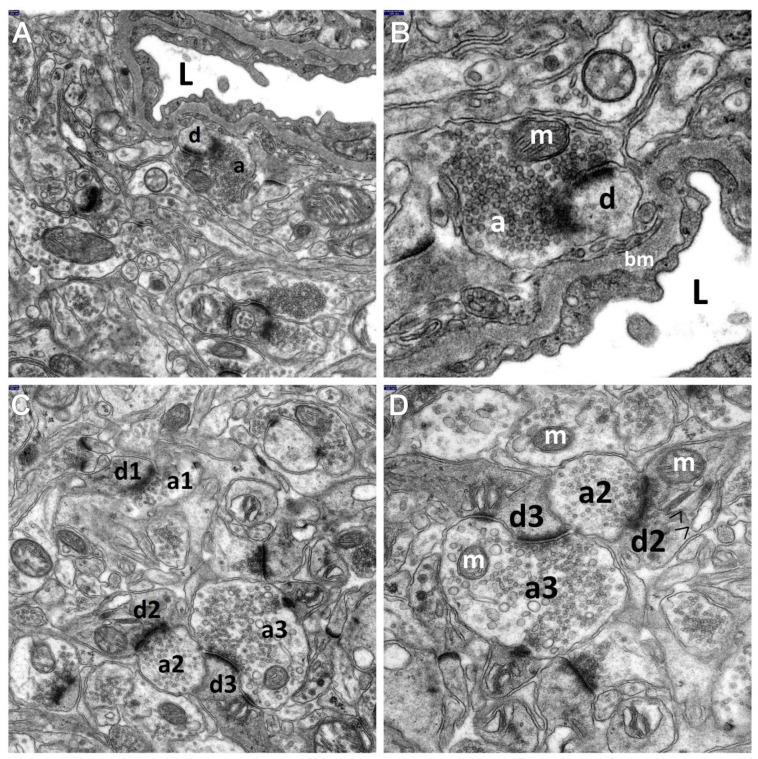
The ultrastructural pictures show the neuroprotective influence of topiramate on the synaptic endings of the neocortex of the temporal lobe in the TPM + HS group. Note that the cytoplasm of these structures is much less swollen as compared to the FS group. (**A**,**B**) Visible is a particularly beneficial effect of the drug on the synaptic endings located directly in the perivascular zone. In a very close vicinity to a quite well-preserved capillary lined with normal endothelial lining, neuropil components can be seen, especially synapses, relatively unchanged or showing slight signs of swelling as compared to the HS group. A greater magnification (**B**) well demonstrates morphological details of the presynaptic (a) and postsynaptic parts of the synapse, (d) directly adhering to the basement membrane (bm) of the capillary. The axonal end-bulb of the presynaptic part contains a large quantity of synaptic vesicles quite evenly distributed within the axoplasm and normal mitochondrion (m); the postsynaptic part (d) is relatively well preserved. Visible are the long active synaptic zone of this axodendritic junction and the well-preserved postsynaptic density thickness. L—perfused capillary lumen. (**C**,**D**) The electonograms clearly demonstrate a favorable effect of TPM on the neuropil components, among which numerous quite well-preserved synaptic junctions are the most visible. Normal picture of the mitochondria (m) present within the cytoplasm of axonal (a) and dendritic (d) end-bulbs. Although some of the axonal-end-bulbs (a2 and a3) are still enlarged, they are filled up with relatively abundant synaptic vesicles evenly distributed within the axoplasm. The dendroplasm of some of the postsynaptic parts of the synapses shows a preserved dilation of the granular endoplasmic reticulum channels (>) (well visible in Figure 4D). Axodendritic junctions show an increase in the length of the synaptic active zone and a relatively well-preserved postsynaptic density. Scale bar: 100 nm (**A**–**D**).

**Figure 5 brainsci-11-01433-f005:**
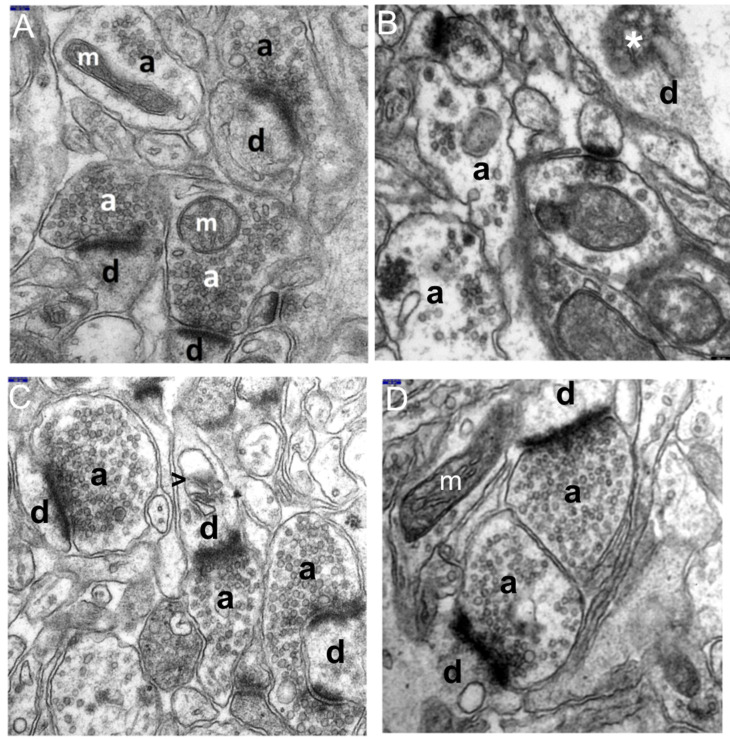
Representative images of EM from different experimental groups (**A**—control group; **B**—FS group; and **C**,**D**—TPM + HS group). All images (**A**–**D**) presented with the same scale bar of 100 nm. (**A**) The elecronogram demonstrates well-preserved neuropil components of the temporal lobe neocortex in the control group. The presynaptic (a) and postsynaptic (d) parts of the synapses are clearly seen, while synaptic active zones are distinct and long, and postsynaptic density was well-preserved (a detailed description below Figure 1). (**B**) The electronogram shows marked swelling and damage to the synaptic endings in the temporal lobe neocortex in the HS group. The presynaptic (a) and postsynaptic (d) parts of synapses are remarkably swollen. Markedly reduced content of synaptic vesicles and their abnormal clumping within the axoplasm can be seen. Within the dendroplasm of dilated synaptic endings, the presence of irregular structures can be seen with a heterogeneous electron density (*) that may correspond to the mitochondrion undergoing necrosis. The length of the synaptic active zones is substantially reduced (a detailed description below Figure 2A,B). (**C**,**D**) Ultrastructural view clearly demonstrating a neuroprotective effect of TPM on the synaptic endings in the TPM + HS group. Although some of the axonal-end-bulbs (a) are still enlarged, they are filled up with relatively abundant synaptic vesicles evenly distributed within the axoplasm. Within the cytoplasm of a single centrally located dendritic ending, dilated channels of smooth endoplasmic reticulum (>) are visible (Figure 5D). However, the majority of the postsynaptic (d) parts of the synapses show normal ultrastructures. Numerous axodendritic junctions show a marked increase in the length of the synaptic active zone and a relatively well-preserved postsynaptic density; normal mitochondrion (m) can be seen in the synaptic ending (Figure 5D).

**Figure 6 brainsci-11-01433-f006:**
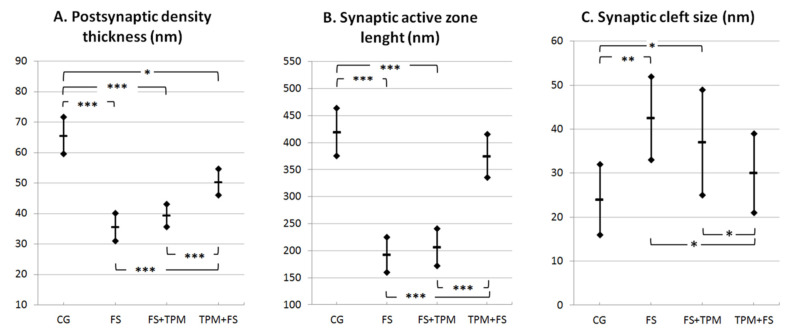
Measurements of postsynaptic density thickness (**A**), synaptic active zone length (**B**), and synaptic cleft size (**C**) in the control group (CG), hyperthermia-induced seizure group (HS), HS group with administrated topiramate (HS + TPM), and group with TPM administration before HS induction (TPM+HS). The median values and interquartile ranges are shown. (**A**) PSD thickness in each group was 65.3 ± 6.1 nm in the CG; 35.5 ± 4.4 nm in HS; 39.2 ± 3.7nm in HS + TPM; and 50.2 ± 4.3 nm in TPM + HS. Measurements significantly differed between the CG and HS (*p* < 0.001), HS + TPM (*p* < 0.001), and TPM + HS (*p* < 0.05) groups, as well as between the TPM + HS and HS (*p* < 0.001), and HS + TPM (*p* < 0.001) groups. No further significant differences were found between any other groups. (**B**) The average synaptic active zone length in each group was 419.3 ± 44.2 nm in the CG; 192.5 ± 32.4 nm in HS; 206.1 ± 34.2 nm in HS + TPM; and 375.9 ± 40.3 nm in TPM + HS. Measurements significantly differed between the CG and HS (*p* < 0.001), and HS + TPM (*p* < 0.001) groups, as well as between the TPM + HS and HS (*p* < 0.001), and HS + TPM (*p* < 0.001) groups, and did not differ significantly between the CG and TPM + FS group (*p* = 0.424). No further significant differences were found between any other groups. (**C**) Synaptic cleft width in each group was 24.3 ± 8.9 nm in the CG; 42.8 ± 10.1 nm in HS; 37.3 ± 12.9 in HS + TPM; and 30.1 ± 9.2 nm in TPM + HS. Measurements significantly differed between the CG and HS (*p* = 0.0014), and HS + TPM (*p* = 0.013) groups, as well as between the TPM + HS and HS (*p* = 0.024), and HS + TPM (*p* = 0.041) groups. No further significant differences were found between any other groups. A threshold significance level: * - *p* ≤ 0.05; ** - *p* ≤ 0.01, *** - *p* ≤ 0.001.

## Data Availability

Not applicable.

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
