# Peer review of "Influence of Topiramate on the Synaptic Endings of the Temporal Lobe Neocortex in an Experimental Model of Hyperthermia-Induced Seizures: An Ultrastructural Study"

_brainsci, 2021, doi:10.3390/brainsci11111433_

Round 1
Reviewer 1 Report
This work analyzed the potentially neuroprotective properties of topiramate against damage to synaptic endings of temporal lobe neocortex in experimental hyperthermia-induced seizures. The manuscript is well written and the findings are of considerable interest. Specific comments are listed below
Materials and methods
- It is not clear why the authors used only male Wistar rats in experiments.
- What region and layer of the temporal lobe neocortex authors used for experiments?
- Please indicated what the statistical analysis you used for comparisons.
Discussion
After reading the discussion section it is not clear the possible of mechanisms of „prophylactically” topiramate actions on the synapse morphology and neurons of neocortex. Please provide more information. For example: In a related study it was reported that TPM modulated the bicarbonate-driven pHi-regulation (Bonnet U, Wiemann M. CNS Neurol Disord Drug Targets. 2020;19(4):264-275. doi: 10.2174/1871527319666200604173208. PMID: 32496992.)
Author Response
Dear Reviewer,
we are very grateful for the time devoted to our manuscript and the in-depth valuable review.
In the beginning, we provide answers to the comments contained in the Material and methods section.
- It is not clear why the authors used only male Wistar rats in experiments.
We decided to use in our experiments only male Wistar rats because in pediatric patients hyperthermia-induced seizures (syn. Febrile seizures) are more common in boys. The male to female ratio is approximately 1.6 to 1.0 (Lueng AK et al. Febrile seizures: an overview. Drugs Context, 2018; in our References Nr. 2). Of course, we added this information in the Material and Methods section. 2.1 in the revised version of our manuscript.
Moreover, it was consistent with the research model, which was a continuation of our previous research.
2. What region and layer of the temporal lobe neocortex authors used for experiments?
We explain that the research material came from the middle of the temporal lobe, which corresponds to gyrus temporalis medius in humans, and mainly from the six-layer (L6)
(layer of polymorphic cells).
3. Please indicated what the statistical analysis you used for comparisons.
Thank you for the question above. In fact, a description of the statistical analysis was missed in the last manuscript.
We used the Kruskal-Wallis H-test, which can be used in such comparisons, being a non-parametric alternative to one-way analysis of variance.
As for the suggestion to extend the section Discussion, to the quotation:
"... the possible of mechanisms of" prophylactically "topiramate actions on the synapse morphology and neurons of neocortex",
in the Discussion section of our revised version of the manuscript, on page 12 we added:
"Recently, it was reported that topiramate may modulate the bicarbonate-driven pHi-regulation [49], which where drug was administered “prophylactically” before experimental HS, may have had a beneficial effect on the synapse morphology and neurons of neocortex."
We also cite the valuable publication suggested by the Reviewer on the mentioned subject (Bonnet U, Wiemann M. Topiramate Decelerates Bicarbonate-Driven Acid-Elimination of Human Neocortical Neurons: Strategic Significance for its Antiepileptic, Antimigraine and Neuroprotective Properties. CNS Neurol Disord Drug Targets. 2020; 19 (4): 264-275. Doi: 10.2174 / 1871527319666200604173208. PMID: 32496992.) (No. 49 in References of the revised version of our manuscript).
Thank you once again for a valuable suggestion that has enriched the substantive value of the work. We hope that the answers provided will be comprehensive. We are grateful for your kindness and the opportunity to cooperate as a reviewer in such a difficult research project.
The noted observation that "The manuscript is well written and the findings are of considerable interest." is motivating and inspiring for us.
With kind regards,
Maria Sobaniec-Lotowska, Prof, MD,
Department of Medical Pathomorphology,
Faculty of Medicine with the Division of Dentistry and Division of Medical Education in English,
Medical University of Bialystok, Poland
maria.sobaniec-lotowska@umb.edu.pl
Reviewer 2 Report
Dear Editor,
The manuscript by Sobaniec et al. investigates the protective effects of topiramate on the temporal lobe neocortex in an experimental model of hyperthermia-induced seizures using EM and ultrastructural level.
The design of the study and the technical quality of the work look somehow convincing and results can be of general interest. However, there is a number of major and minor points that would need to be addressed in order to improve the quality of this paper before it can be considered for publication:
General:
-The last bit of the title is confusing and it needs to be deleted.
-It was to be much easier if the manuscript includes line numbers.
-Defining abbreviation whenever they are firstly introduced and keep using them throughout the manuscript. For example: FS in line 5, CNS, topiramate (as in the abstract) of the introduction and others throughout the manuscript.
-The manuscript needs careful proofreading. For example, syn. In line 2 of the introduction.
Major:
-I strongly discourage the use of primacy and novelty claims (e.g., new, novel, the first, etc). Please revise the full manuscript accordingly, including the title.
-Authors need to perform a post hoc correction to moderate the variability in comparing multiple samples. Bonferroni correction or Conover-Inman post hoc corrections must be performed. Authors need to provide the new corrected p values and report any difference in their observations.
-The authors have not conclusively demonstrated the effect of topiramate for the following reasons:
- There was no indication for the time-dependent effect of the drug. How long is its duration of action and half-life? Without this information, it’s hard to believe that the effect is only prophylactic since the indicated 90 minutes for the drug administration is most probably within the normal duration of its action.
- The study only includes EM data without any functional or behavioural experiments to make sense of the seen phenotype and structural observations.
- The manuscript failed to answer a simple question which is whether before or after topiramate administration was associated with less episodes or duration of convulsions before correlating this to the ultrastructural findings. Without having less seizure, no ultrastructural data will make any sense.
-There are numerous claims throughout the manuscript that are inaccurate or poorly supported by the up-to-date literature. For example: “The neuroprotective effect of topiramate on the ultrastructural picture of the synapses was demonstrated mainly by less substantial swelling of the synaptic endings”. This is most probably due to the effect of the drug at the astrocytes in the tripartite synapse where AQP4 have been shown to play a major role in swelling and the pathophysiology of epilepsy. The Discussion lacks a brief mention regarding the essential role of glial cells. Glial cells, particularly astrocytes, appear to play critical and interactive roles swelling and functional recovery. A recent breakthrough work by Kitchen et al Cell 2020, has showed that they are important targets for CNS disorders. These results have been confirmed by the work of Sylvain et al. who confirmed the role in stroke. References to be included:
- https://pubmed.ncbi.nlm.nih.gov/32413299/
https://pubmed.ncbi.nlm.nih.gov/33561476/
The role of AQP4 in epilepsy is well-stablished. Authors need to discuss their findings in the light of these studies:
https://pubmed.ncbi.nlm.nih.gov/28715131/
https://pubmed.ncbi.nlm.nih.gov/22378467/
-Discussion “The protective effect of topiramate on the BBB components, mainly on the endothelial lining of capillaries has been confirmed” and then “results in substantial enhancement of the BBB and thus better blood supply in the CNS”. The stabilization of BBB has recently been shown to improve the glymphatic function and hence functional recovery after condition like epilepsy. References to be included:
https://pubmed.ncbi.nlm.nih.gov/34499128/
https://pubmed.ncbi.nlm.nih.gov/34408336/
Minor:
-Discussion “Taking into consideration several times faster metabolism of rats compared to that of humans, a dose of 80 mg/kg b.m. of TPM, in our opinion, is safe and comparable to the dose administered e.g. to patients with drug-resistant epilepsy. Possibly, the use of a higher dose than 80 mg/kg b.m. might ensure more beneficial and at the same time safe neuroprotective effect of the drug on the synaptic endings of the CNS structures studied. This, however, would require further neuropharmacological research extended with ultrastructural investigations”. This paragraph needs to be moved to the materials and methods.
-Conclusion “i.e. through the so called vascular factor”. This’s vague, elaborate.
-Introduction “(i.e. qualitative and quantitative investigations)”. This isn’t needed and better be deleted.
Best
Author Response
Dear Reviewer,
Once again, we would like to thank you for your time, for a thorough evaluation, valuable comments, and important suggestions on our manuscript.
In such a short time, we tried to answer most of the questions discussed.
-The last bit of the title is confusing and it needs to be deleted.
-I strongly discourage the use of primacy and novelty claims (e.g., new, novel, the first, etc). Please revise the full manuscript accordingly, including the title.
Following the first remark that in the full manuscript, including the title, we should not use the terms "new, novel, the first, etc", we included this suggestion in the new version of our work by updating the title and the full manuscript.
The current title of our manuscript is as follows: "Influence of topiramate on the synaptic endings of the temporal lobe neocortex in an experimental model of hyperthermia-induced seizures - an ultrastructural study"
In the new version of the work, we also tried to take into account the other comments contained in the General criticism section.
-It was to be much easier if the manuscript includes line numbers.
The manuscript uploaded by the authors is automatically converted into pdf, to which the line numbers are added. Due to technical problems, we were unable to send the manuscript, so the last version of the manscript was sent by the MPDI editor - with no lines numbers.
-Authors need to perform a post hoc correction to moderate the variability in comparing multiple samples. Bonferroni correction or Conover-Inman post hoc corrections must be performed. Authors need to provide the new corrected p values and report any difference in their observations.
Thank you for your suggestion on statistical analysis. Indeed, a detailed description of the statistical analysis was missed in the last manuscript which is corrected. In our study, we used the Kruskal-Wallis H-test method, which can also be used in such comparisons, as a test that is a non-parametric alternative to one-way analysis of variance. With this test, we compared the distributions of the variables. After consulting our statistician, we decided to keep the current statistical analysis and the methods used.
-The authors have not conclusively demonstrated the effect of topiramate for the following reasons
Regarding the comment in Major: "The study only includes EM data without any functional or behavioral experiments to make sense of the seen phenotype and structural observations. "
Well, we want to note that the purpose of our current morphological research was to estimate potentially neuroprotective effects of TPM against submicroscopic damage to the synaptic endings in the temporal lobe neocortex caused by hyperthermic convulsions in rats using EM (including quality descriptive changes and morphometric analysis).
Here we agree with the Reviewer who, referring to the lack of functional tests, writes: "The manuscript failed to answer a simple question which is whether before or after topiramate administration was associated with less episodes or duration of convulsions before correlating this to the ultrastructural findings ".
We would like to explain that we have been conducting experimental research on the influence of TPM on the CNS in febrile seizures in rats at our Center for many years (Ref. no. 13,22,28,29, 34-38) and it is obvious that from the beginning, i.e. before the application of morphological tests carried out with the use of light microscopy (LM) and transmission electron microscopy (TEM), we preceded functional and behavioral observations of animals in individual research groups combined with preliminary microscopic examination of selected structures CNS.
On their basis, we found that "prophylactically" the administration of TPM, i.e. before hyperthermic-induced seizures (TPM + HS group), resulted in a reduction of episodes or duration of seizures, which correlated with the preliminary morphological findings. On the other hand, the functional state of animals did not change when topiramate was applied after hyperthermic-induced seizures, i.e. in the HS + TPM group.
Thank you for the suggestion to add functional or behavioral experiments with TPM, which we will take into account in future AEDs research, also TPM.
Extending the content of the present work with a panel of precise functional observations of functional observations would result in a complete reconstruction of the manuscript and does not concern the purpose of the work, which is electron microscopy evaluation and analysis.
-The Discussion lacks a brief mention regarding the essential role of glial cells. Glial cells, particularly astrocytes, appear to play critical and interactive roles swelling and functional recovery. A recent breakthrough work by Kitchen et al Cell 2020, has showed that they are important targets for CNS disorders. These results have been confirmed by the work of Sylvain et al. who confirmed the role in stroke.
Regarding the comments Reviewer regarding the essential role of the population of glial cells, particularly astrocytes, in our experiment with TPM which "appear to play critical and interactive roles swelling and functional recovery."
We agree with this statement, especially as it is supported by the results of our previous extensive and laborious observations on the electron microscopy image of astrocytes of the cortex of the hippocampal gyrus as well as in the neocortex of the temporal lobe in an analogous experimental model of HS and with the use of topiramate.
Łotowska JM , Sobaniec-Łotowska ME, Sobaniec W. Ultrastructural features of astrocytes in the cortex of the hippocampal gyrus and in the neocortex of the temporal lobe in an experimental model of febrile seizures and with the use of topiramate. Folia Neuropathol. 2009;47(3):268-77.
We have already quoted the above work in the introduction to our manuscript (No. 36 in the References), but as expected of the Reviewer we refer to it now in the Discussion of the revised manuscript.
"Interestingly, in our earlier submicroscopic study in an analogous experimental model of hyperthermia-induced seizures, when TPM was applied prior to induction of HS, the drug was found to have a beneficial effect on the structural status of approximately 1/3 of the population of protoplasmic astroglial cells of the hippocampal cortex and neocortex of the temporal lobe which may be explained, among others, by a protective action of cerebral cortex microcirculation [35].
It is worth adding that in the light of recent research, this is most probably due to the effect of the drug at the astrocytes in the tripartite synapse where aquaporin 4 (AQP4) channels, which are mainly expressed in perivascular astrocyte endfeet, have been shown to play a major role in cell swelling and the pathophysiology of epilepsy [45, 46].
The population of glial cells, particularly astrocytes, appear to play critical and interactive roles swelling and functional recovery [46-48] and are important targets for CNS disorders [46, 48]."
The above fragment is added to the Discussion and extended with references proposed by the reviewer:
Binder DK, Nagelhus EA, Ottersen OP. Aquaporin-4 and epilepsy. Glia. 2012 Aug;60(8):1203-14. doi: 10.1002/glia.22317. Epub 2012 Feb 29. PMID: 22378467
Kitchen P, Salman MM , Halsey AM et al. Targeting Aquaporin-4 Subcellular Localization to Treat Central Nervous System Edema. Cell. 2020 May 14;181(4):784-799.e19. doi: 10.1016/j.cell.2020.03.037.
Salman MM, Kitchen P, Halsey A, Wang MX, Tornroth-Horsefield S, Conner AC, Badaut J, Iliff JJ, Bill RM. Emerging roles for dynamic aquaporin-4 subcellular relocalization in CNS water homeostasis. Brain. 2021 Sep 9:awab311. doi: 10.1093/brain/awab311. Online ahead of print.
-Discussion “The protective effect of topiramate on the BBB components, mainly on the endothelial lining of capillaries has been confirmed” and then “results in substantial enhancement of the BBB and thus better blood supply in the CNS”. The stabilization of BBB has recently been shown to improve the glymphatic function and hence functional recovery after condition like epilepsy
We thank the Reviewer for the interesting statement that “The stabilization of BBB has recently been shown to improve the glymphatic function and hence functional recovery after condition like epilepsy”.
In our opinion, it is highly probable that an analogous or similar stabilization mechanism of the BBB could also occur in our animal model of hyperthermic and TPM seizures, which, requires further in-depth research.
We extend the Discussion of our revised manuscript with the fragment mentioned above, and we add two references proposed by Reviewer:
Salman MM, Kitchen P, IIiff JJ, Bill R M. Aquaporin 4 and glymphatic flow have central roles in brain fluid homeostasis. Nature Revies/Neurosciences, Correspondence, 2021…???
Salman MM, Kitchen P, Halsey A, Wang MX, Tornroth-Horsefield S, Conner AC, Badaut J, Iliff JJ, Bill RM. Emerging roles for dynamic aquaporin-4 subcellular relocalization in CNS water homeostasis. Brain. 2021 Sep 9:awab311. doi: 10.1093/brain/awab311. Online ahead of print.
As suggested by the Reviewer, we moved from Discussion section to the Material and methods section of revised version manuscript the following paragraph ”Taking into consideration several times faster metabolism of rats compared to that of humans, a dose of 80 mg / kg b.m. of TPM with ultrastructural investigations ”.
-Conclusion “i.e. through the so called vascular factor”. This’s vague, elaborate.
We also omitted the phrase “i.e. through the so called vascular factor ”.
-Introduction “(i.e. qualitative and quantitative investigations)”. This isn’t needed and better be deleted.
Moreover, as suggested by the Reviewer, we deleted in Introduction of revised version of our manuscript from the term “i.e.qualitaive and quantitative investigations”.
Thank you once again for a valuable suggestion that has enriched the substantive value of the work. We hope that the answers provided will be comprehensive. We are grateful for your kindness and the opportunity to cooperate as a reviewer in such a difficult research project.
The noted observation that "The design of the study and the technical quality of the work look somehow convincing and results can be of general interest" is motivating and inspiring for us.
With kind regards,
Maria Sobaniec-Lotowska, Prof, MD,
Department of Medical Pathomorphology,
Faculty of Medicine with the Division of Dentistry and Division of Medical Education in English,
Medical University of Bialystok, Poland
maria.sobaniec-lotowska@umb.edu.pl
Reviewer 3 Report
The manuscript "Influence of topiramate on the synaptic endings of the temporal lobe neocortex in an experimental model of hyperthermia-induced seizures. The first ultrastructural study” by Drs. Piotr Sobaniec et al is aimed to explore the potential neuroprotector topyramate (TPM). TPM was administered to rats before and immediately after epileptic seizures elicited by hyperthermia (HS). After that, the samples of brain tissue were examined using transmission electron microscopy. The authors concluded that preventive administration of TPM prior to HS induction demonstrates neuroprotective TPM property.
I have some question:
It was shown (Anna Andreou, Peter Goadsby, Cephalalgia 2011 Oct;31(13):1343-58. doi: 10.1177/0333102411418259) that topiramate modulates trigeminovascular transmission within the trigeminothalamic pathway with the glutamate-kainate receptor.
Do the authors believe that similar mechanisms were involved in their experiments?
Minor criticism:
The scale bars in all figures are very small, and the numbers are not visible.
The data is original and impressive, the manuscript organized well and written clear. I am happy to recommend the manuscript for the publication after minor corrections, suggested before.
Author Response
Dear Reviewer,
we are very grateful for the time devoted to our manuscript and the in-depth valuable review.
It was shown (Anna Andreou, Peter Goadsby, Cephalalgia 2011 Oct;31(13):1343-58. doi: 10.1177/0333102411418259) that topiramate modulates trigeminovascular transmission within the trigeminothalamic pathway with the glutamate-kainate receptor.
Do the authors believe that similar mechanisms were involved in their experiments?
Thank you for your comments and suggestions on our manuscript on the potential neuroprotective properties of TPM against damage to synaptic endings of the temporal lobe neocortex in experimental hyperthermia-induced seizures.
Thank you very much for drawing your observation that:
"topiramate modulates trigeminovascular transmission within the trigeminothalamic pathway with the glutamate-kainate receptor contained in the work by Anna Andreou and Peter Goadsby in Cephalalgia 2011 Oct; 31 ( 13): 1343-58. doi: 10.1177 / 0333102411418259).
However, answering question Reviewer 3 to the possible mechanisms of TPM operation, we believe that similar mechanisms may have been involved in our experiments, but it requires further research. We cite the above item in the references of our revised manuscript in the section Introduction: (new References No. 25).
The scale bars in all figures are very small, and the numbers are not visible.
The figures sent to the Editorial Office were of better quality - higher resolution than the figures included in the last manuscript. It seems to us that they might have been converted worse at the stage of creating the final file by the system.
Moreover, the details are additionally included in the captions under the figures.
Thank you once again for suggestions that have enriched the substantive value of the work. We hope that the answers provided will be comprehensive. We are grateful for your kindness and the opportunity to cooperate as a reviewer in such a difficult research project.
The noted observation that "The data is original and impressive, the manuscript organized well and written clear. " is motivating and inspiring for us, especially qualitative and quantitative ultrastructural research on TPM and other antiepileptic drugs.
With kind regards,
Maria Sobaniec-Lotowska, Prof, MD,
Department of Medical Pathomorphology,
Faculty of Medicine with the Division of Dentistry and Division of Medical Education in English,
Medical University of Bialystok, Poland
maria.sobaniec-lotowska@umb.edu.pl
Round 2
Reviewer 2 Report
Dear Editor,
The authors have successfully addressed the majority of my comments and concerns in order to improve the quality of the manuscript.
I believe that the new section, improved methodology, and updated references, have contributed to enhancing the clarity of the manuscript, which I can now endorse for publication.
All the best!
This manuscript is a resubmission of an earlier submission. The following is a list of the peer review reports and author responses from that submission.
Round 1
Reviewer 1 Report
The manuscript by Piotr Sobaniec and colleagues is an interesting but limited and descriptive study of the potential beneficial effects of Topiramate administration to prevent synaptic alterations induced by hyperthermia-induced seizures in rodents. I consider that the manuscript presents major flaws and limitations that should be addressed before being acceptable for publication. The lack of morphometric analysis (i.e. quantitative studies) of morphological changes and absence of appropriate statistical analysis are also unacceptable for a modern morphological study and should be performed to validate these results and the significance of the observed changes. In detail:
Major comments:
- The authors use the term “febrile seizures” as a synonymous of “hyperthermia-induced seizures”. That is not correct because febrile seizures are usually caused by bacterial infection where LPS and other PAMP are circulating and thus innate immune system is activated with cytokine secretion that, even with intact BBB, activate microglial cells and astrocytes in the CNS. On the other hand, hyperthermia-induced seizures lacks of the intense immune system activation induced by bacterial LPS or other PAMPs. Thus, extremely care should be taken before translating into potential clinical settings these results. Please introduce a “warning sign” in the discussion about this issue and do not interchange the concepts along the manuscript. As an example please see Feng and Chen (2016), Neurosci Bull. 2016 Oct;32(5):481-92.
- Please explain why the authors have decided to use only male rats and the reasons that justify the use of 22-30 days old rats (that already have reached the ability of regulate the body temperature). Most authors use 10-days old rats in hyperthermia-induced seizures. Please explain the reasons that justify the use of such age.
- Do the authors “dry” the rats or do any other treatment to prevent animal behavioural alterations due to the exposure to hot water during the 4-days of treatment? I am worry about the stress, behavioural changes and feeding behaviour of these water-exposed animals. Are control animals being exposed to room temperature water for the same time during the 4-days? Please clarify.
- One of the main issues in results section is that changes in morphology are not quantified. This is unacceptable in modern morphological sciences. In addition, TEM images (that are in excellent in quality) are not identified with arrows, asterisks, etc, to make the images understandable for any reader interested in epilepsy. Please consider that this potential paper should have interest for a very broad scientific and medical audience. Please indicate main features in images and perform quantitative studies of the changes observed, with statistical analysis and “n” analyzed.
- Figure legends refer to markers that are not present in the images (ie. “ the presynaptic (a1, a2, a3); postsynaptic (d1, d2, d3); the asterisk is absent in Figure 2D, etc). Please revise and whenever is possible use arrows, arrow heads, etc that are easier to follow.
- Also in Figure 2, authors describe that synaptic cleft is expanded, but no quantitative measurements are presented. Again, this is unacceptable for modern studies on CNS morphology.
- Authors propose that changes may be due to apoptosis or necrosis. Please try to discriminate that, for example by analyzing caspases activity or by improving the analysis of TEM images searching for apoptotic cell bodies, membranes disintegration, etc. The enormous interest of this analysis is that apoptosis occurs without noticeable neuroinflammation, while necrosis produce a large neuroinflammatory stimulus with massive release of DAMPs and activation of the CNS resident innate immunity system (microglia, astrocytes and perivascular macrophages). If topiramate is able to prevent also that effect, that could be an extremely interesting beneficial side effect of the drug.
- Figure 3 and 4 also lacks of morphometrical analysis and statistical description.
- The results in figure 4 show that pre-treatment with topiramate seem to be beneficial. I am trying to imagine the potential clinical use of such condition. Because the clinician should administrate topiramate in a preventive form? Before the onset of seizures?. Please discuss how authors propose that these results could be translated into clinically relevant findings.
Author Response
Dear Reviewer R1,
Enclosed please find our revised manuscript with the changes suggested in the previous review entitled: 'Influence of topiramate on the synaptic endings of the temporal lobe neocortex in an experimental model of hyperthermia-induced seizures. The first ultrastructural study ' by Piotr Sobaniec, Joanna M. Lotowska, Maria E. Sobaniec-Lotowska, Milena Zochowska-Sobaniec from Department of Pediatric Neurology and Rehabilitation, Medical University of Bialystok, Poland, Department of Medical Pathomorphology, Medical University of Bialystok, Poland, and Department of Pediatrics, Gastroenterology, Hepatology, Nutrition and Allergology, Medical University of Bialystok, Poland, with a great request to publish it in Brain Sciences.
We are very grateful to Reviewer R1 for the time devoted to our manuscript and the in-depth valuable review.
Referring to point 1:
- The authors use the term “febrile seizures” as a synonymous of “hyperthermia-induced seizures”. That is not correct because febrile seizures are usually caused by bacterial infection where LPS and other PAMP are circulating and thus innate immune system is activated with cytokine secretion that, even with intact BBB, activate microglial cells and astrocytes in the CNS. On the other hand, hyperthermia-induced seizures lacks of the intense immune system activation induced by bacterial LPS or other PAMPs. Thus, extremely care should be taken before translating into potential clinical settings these results. Please introduce a “warning sign” in the discussion about this issue and do not interchange the concepts along the manuscript. As an example please see Feng and Chen (2016), Neurosci Bull. 2016 Oct;32(5):481-92.
As suggested by R1, after the analysis of the suggested publications in the revised manuscript we have changed the term “febrile seizures” for “hyperthermia-induced seizures”. We agree with R1 that ”…because febrile seizures are usually caused by bacterial infection where LPS and other PAMP are circulating and thus the immune system is activated with cytokine secretion …”. We are very grateful for the suggestion to take advantage of the valuable publication of Feng and Chen (2016).
Feng, B. and Z. Chen, Generation of Febrile Seizures and Subsequent Epileptogenesis. Neurosci Bull, 2016. 32(5): p. 481-92.
We have also added this publication to the References of our revised manuscript.
Point 2:
- Please explain why the authors have decided to use only male rats and the reasons that justify the use of 22-30 days old rats (that already have reached the ability of regulate the body temperature). Most authors use 10-days old rats in hyperthermia-induced seizures. Please explain the reasons that justify the use of such age.
Referring to the question contained in point 2, we would like to explain that we have decided to use only male rats, as this condition, i.e. febrile seizures, is more common in boys – the male to female ratio is approximately 1.6 to 1.0.
Leung, A.K., K.L. Hon, and T.N. Leung, Febrile seizures: an overview. Drugs Context, 2018. 7: p. 212536.
Referring to the reason that justifies the use of 22-30 days old rats, taking into account that these seizure disorders occur in children in a wide time interval, i.e. mainly between 6 months and 5 years, in pediatric patients, which is connected with a varied degree of structural brain maturation, we decided to use animals older than 10-days, (i.e. 22-30 days). In the manuscript Sample et al. (2013) concerning brain development in rodents and humans, containing a detailed sum-up table (Tab. 1) of key fundamental brain developmental processes across comparable ages in humans and rodents, brain maturity in rodents aged 20-21 pnd corresponds to that in humans aged 2-3 years, and in rats aged 25-35 pnd to brain maturity in 4-11-year-old humans (after Tsujimoto 2008)
Semple, B.D., et al., Brain development in rodents and humans: Identifying benchmarks of maturation and vulnerability to injury across species. Prog Neurobiol, 2013. 106-107: p. 1-16.
Tsujimoto, S., The prefrontal cortex: functional neural development during early childhood. Neuroscientist, 2008. 14(4): p. 345-58.
Point 3:
- Do the authors “dry” the rats or do any other treatment to prevent animal behavioural alterations due to the exposure to hot water during the 4-days of treatment? I am worry about the stress, behavioural changes and feeding behaviour of these water-exposed animals. Are control animals being exposed to room temperature water for the same time during the 4-days? Please clarify.
Referring to point 3 (if we understood well).
The animals after being taken out of hot water were not specially “dry” – they were moved to a separate container lined with lignin.
On the other hand, control animals were not exposed to room temperature water for the same time during the 4-day period.
Points 4, 5, 6:
- One of the main issues in results section is that changes in morphology are not quantified. This is unacceptable in modern morphological sciences. In addition, TEM images (that are in excellent in quality) are not identified with arrows, asterisks, etc, to make the images understandable for any reader interested in epilepsy. Please consider that this potential paper should have interest for a very broad scientific and medical audience. Please indicate main features in images and perform quantitative studies of the changes observed, with statistical analysis and “n” analyzed.
- Figure legends refer to markers that are not present in the images (ie. “ the presynaptic (a1, a2, a3); postsynaptic (d1, d2, d3); the asterisk is absent in Figure 2D, etc). Please revise and whenever is possible use arrows, arrow heads, etc that are easier to follow.
- Also in Figure 2, authors describe that synaptic cleft is expanded, but no quantitative measurements are presented. Again, this is unacceptable for modern studies on CNS morphology.
As Reviewer R1 made some major reservations referring to the lack of morphometric analysis of morphological changes within synapses and absence of appropriate statistical analysis we have extended the text adding the quantitative investigations. Therefore, the volume of the revised manuscript is ¼ bigger as compared to the previous version.
We would like to add that due to the fact that this is the first morphological analysis of this drug in experimental neurology, referring to morphological changes in the synaptic endings of the temporal lobe neocortex following hyperthermia-induced seizures, with TPM administration, the interpretation difficulty level is very high as our findings cannot be compared to observations made by other authors. However, we hope that the currently revised manuscript (with added morphometric analysis and statistics) will meet the expectations of the Reviewers, being acceptable for a modern morphological study.
We are grateful to R1 for valuable technical comments in points 4 and 5. In the revised manuscript, we have added appropriate graphic symbols as suggested (e.g. the asterisk in Figure 2D; missing a1, a2, a3…).
We are also grateful to R1 for the statement in point 4 that TEM images are excellent in quality, which encourages us to improve the whole manuscript and to continue our further morphological studies on the drug in future research.
Referring to the comments in points 4, 6, 7:
- Authors propose that changes may be due to apoptosis or necrosis. Please try to discriminate that, for example by analyzing caspases activity or by improving the analysis of TEM images searching for apoptotic cell bodies, membranes disintegration, etc. The enormous interest of this analysis is that apoptosis occurs without noticeable neuroinflammation, while necrosis produce a large neuroinflammatory stimulus with massive release of DAMPs and activation of the CNS resident innate immunity system (microglia, astrocytes and perivascular macrophages). If topiramate is able to prevent also that effect, that could be an extremely interesting beneficial side effect of the drug.
- Figure 3 and 4 also lacks of morphometrical analysis and statistical description.
Since major reservations referring to the lack of morphometric analysis of morphological changes within synapses and the absence of appropriate statistical analysis we have extended the text adding the quantitative investigations in Fig. 6, including characteristics, description and commentary. Therefore, the volume of the completed manuscript is ¼ bigger as compared to the previous version.
In point 7 R1 writes that “Authors propose that changes may be due to apoptosis or necrosis “ and considers the possibility of their biochemical (enzymatic) or morphological identification.
Indeed, the cytoplasm of some degenerated synaptic endings, especially in their postsynaptic parts, showed fine, interesting changes manifested as irregularly shaped heterogeneous electron-dense structures of mixed electron density, whose exact origin we could not determine.
We state in our manuscript: “Such structures were observed in endoplasmic reticulum, among cytoskeletal elements and in the mitochondrial matrix of some damaged and disrupted mitochondria. They may constitute the morphological feature of necrotic changes or apoptosis corresponding to neuronal cell death due to hyperthermic stress (Figure 2A, B, D).” However, we add that ” the exact identification of such structures with mixed electron density, sometimes observed within the synapses, requires further in-depth studies, including the immunological ones”. As this is the first morphological work in neuropathology on the effect of TPM on the synaptic endings of the neocortex (signaling the morphogenesis of these changes), we hope to continue our research and that other author from other Centers will get involved in the subject.
Of great help would be the ultrastructural studies conducted by authors from different Centers concerned with the effects of antiepileptics on the microscopic-electron picture of the CNS.
Thus, our results of submicroscopic analysis of synaptic endings, at this stage of observation, may in the future serve as an interesting comparative material for similar studies.
We put great hope in the results of studies conducted by other Centers working on the ultrastructural evaluation of the CNS.
Referring to point 8:
- The results in figure 4 show that pre-treatment with topiramate seem to be beneficial. I am trying to imagine the potential clinical use of such condition. Because the clinician should administrate topiramate in a preventive form? Before the onset of seizures?. Please discuss how authors propose that these results could be translated into clinically relevant findings.
We are grateful to R1 for the valuable comment and at the same time a broad view of the potential application of TPM in clinical practice. R1 writes: “The results in Figure 4 showing that pre-treatment with topiramate seems to be beneficial” and then adds “I am trying to imagine the potential clinical use of such condition.”
TPM has been suggested to exhibit neuroprotective and antiepileptic effects and was therefore used in this study.
Edmonds HL Jr, Jiang YD, Zhang PY and Shank R: Topiramate as a neuroprotectant in a rat model of global ischemia-induced neurodegeneration. Life Sci. 69:2265–2277. 2001.
We believe that this report is of interest to the potential Readers of Brain Sciences neither only for neurologists nor neuropathologists but also clinicians. Thanking the Editor and potential Reviewers for their efforts I remain with best regards.
We hope that our explanations will be satisfactory to the Editors and specially Reviewer R1 and that our manuscript will be suitable for publication in Brain Sciences.
We look forward to hearing from you soon.
Yours sincerely
Maria E. Sobaniec-Lotowska, Prof. MD, PhD,
Department of Medical Pathomorphology,
Medical University of Bialystok, Poland;

Reviewer 2 Report
In this manuscript, the authors investigated the ultrastructural changes in the synaptic endings of temporal lobe neocortex following hyperthermia-induced seizures in the water bath model, as well as the effects of topiramate. They found that topiramate given before FS could reverse the damages in synaptic endings, including synaptic swelling, damaged mitochondria, decreased numbers of vescicles in the presynaptic boutons, and shortened active zone , which were present in FS group and FS group with topiramate given immediately after FS occurrence.
Here are my comments.
- Methods lack the details. For example, there is no description of control goup and EM procedure.
- There are no quantifications and statistical analysis for all data presented.
- Representative images of EM from different groups should be put together with the same scale for better comparisons.
- There is no description of the seizures induced by hyperthermia in the results, such as duration, latency, severity, and percentage of animals that had seizures. Does application of topiramate before FS affect seizures?
- What is the significance of the study on the synaptic endings following hyperthermia-induced seizures? Does the synaptic damage following hyperthermia-induced seizures have certain correlation with later chronic epilepsy, or other behavioral deficits? Were those changes in synaptic endings due to hyperthermia, convulsions, or stress? These are some of the questions that would be interesting to discuss in the manuscript.
Author Response
Dear Reviewer R2,
Enclosed please find our revised manuscript with the changes suggested in the previous review entitled: 'Influence of topiramate on the synaptic endings of the temporal lobe neocortex in an experimental model of hyperthermia-induced seizures. The first ultrastructural study' by Piotr Sobaniec, Joanna M. Lotowska, Maria E. Sobaniec-Lotowska, Milena Zochowska-Sobaniec from Department of Pediatric Neurology and Rehabilitation, Medical University of Bialystok, Poland, Department of Medical Pathomorphology, Medical University of Bialystok, Poland, and Department of Pediatrics, Gastroenterology, Hepatology, Nutrition and Allergology, Medical University of Bialystok, Poland, with a great request to publish it in Brain Sciences.
We are very grateful to Reviewer R2 for the time devoted to our manuscript and the in-depth valuable review and patience.
We would like to add that due to the fact that this is the first morphological analysis of this drug in experimental neurology, referring to morphological changes in the synaptic endings of the temporal lobe neocortex following hyperthermia-induced seizures, with TPM administration, the interpretation difficulty level is very high as our findings cannot be compared to observations made by other authors. However, we hope that the currently revised manuscript (with added morphometric analysis and statistics) will meet the expectations of the Reviewer, being acceptable for a modern morphological study.
Referring to point 1:
- Methods lack the details. For example, there is no description of control goup and EM procedure.
Referring to point 1: “Methods lack the details” we have completed the methodology in the control group in Material and Methods adding that this group received only normal saline in an amount of 2 ml with an intragastric tube.
The EM procedure in the control group is the same as in the experimental groups, which is described in point 2.3 Preparation for transmission electron microscopy (TEM) (page 3; lines 109-124 or 123-124 ): “The material obtained from the neocortex of temporal lobe in the control group was processed…”
Details of the morphological view of neuropil components of the temporal lobe neocortex in the control group examined in TEM demonstrate the electronogram 1 (Fig. 1), including a detailed description placed below the figure. (page 4).
- There are no quantifications and statistical analysis for all data presented.
Taking into account the comment: ”There no quantifications and statistical analysis for all data presented” we have extended the text of the revised manuscript adding quantitative investigations, contained in Figure 6, including their characteristics, description and commentary. We agree with R2 that such investigations should be conducted to assess potentially neuroprotective properties of TPM with reference to the neocortical synaptic endings studied.
We hope that the morphometric analysis (i.e. quantitative investigations) and appropriate statystical analysis for all data presented have allowed us to objectivize the results of descriptive analysis referring to the synaptic endings.
Therefore the volume of the current version is approximately by ¼ bigger as compared to the previous one.
- Representative images of EM from different groups should be put together with the same scale for better comparisons.
According to the suggestion of R2, we have added representative images of EM from different experimental groups (control group, FS group and TPM+FS group), which were put together with the same scale (scale bar 100 nm) for better comparisons ( Figure 5 A-D; page 9). This helped to present differences between the analyzed parameters in the groups.
- There is no description of the seizures induced by hyperthermia in the results, such as duration, latency, severity, and percentage of animals that had seizures. Does application of topiramate before FS affect seizures?
Referring to point 4, we would like to explain that the revised manuscript has been extended by adding the descriptive characteristics of the seizures induced by hyperthermia in our animals.
Most rats subjected to hyperthermia hot water developed rapidly myoclonic jerks and then generalized seizures, with vigorous shaking of the head, ears, and upper and lower limbs and an especially violent vibration of the tail.
We included this fragment in point 2.2. Model of febrile seizures (page 3) of the new manuscript.
However, we observed that the application of TPM before FS affect seizures decreased the intensity, amplitude and duration of the seizures, which was the subject of our earlier conference report (Sendrowski 2012, KUL, Poland).
We have not conducted a statistical analysis of the respective symptoms.
In the future, when we repeat our experiment, we are planning to extend our investigations with quantitative observations, such as duration, latency, severity and percentage of animals that had seizures.
- What is the significance of the study on the synaptic endings following hyperthermia-induced seizures? Does the synaptic damage following hyperthermia-induced seizures have certain correlation with later chronic epilepsy, or other behavioral deficits? Were those changes in synaptic endings due to hyperthermia, convulsions, or stress? These are some of the questions that would be interesting to discuss in the manuscript.
We are very grateful for the interesting and at the same time difficult questions in point 5. Based on our own observations and literature review we can guess that the observed structural changes in the synaptic endings were caused by the combined action of hyperthermia and convulsions, with possible involvement of stress. In our opinion, hyperthermia had the greatest impact, which is consistent with the reports described. We included this statement in our Discussion.
A possible correlation of the synaptic damage with later chronic epilepsy or other behavioral deficits were not part of this study.
We believe that this report is of interest to the potential Readers of Brain Sciences neither only for neurologists nor neuropathologists but also clinicians. Thanking the Editor and Reviewer R2 for their efforts I remain with best regards.
We hope that our explanations will be satisfactory to the Editors and specially Reviewer R2 and that our manuscript will be suitable for publication in Brain Sciences.
We look forward to hearing from you soon.
Yours sincerely
Maria E. Sobaniec-Lotowska, Prof. MD, PhD,
Department of Medical Pathomorphology,
Medical University of Bialystok, Poland;
